# Evaluations of Andrographolide-Rich Fractions of *Andrographis paniculata* with Enhanced Potential Antioxidant, Anticancer, Antihypertensive, and Anti-Inflammatory Activities

**DOI:** 10.3390/plants12061220

**Published:** 2023-03-07

**Authors:** Sya’ban Putra Adiguna, Jonathan Ardhianto Panggabean, Respati Tri Swasono, Siti Irma Rahmawati, Fauzia Izzati, Asep Bayu, Masteria Yunovilsa Putra, Carmen Formisano, Chianese Giuseppina

**Affiliations:** 1Department of Chemistry, Faculty of Mathematics and Natural Sciences, Universitas Gadjah Mada, Bulaksumur, Yogyakarta 55281, Indonesia; syaban.putra@mail.ugm.ac.id (S.P.A.); jonathanpanggabean@mail.ugm.ac.id (J.A.P.); respati@ugm.ac.id (R.T.S.); 2Research Center for Vaccine and Drugs, Research Organisation for Healths, National Research and Innovation Agency (BRIN), Jalan Raya Jakarta-Bogor KM. 46, Cibinong 16911, Indonesia; fauz010@brin.go.id (F.I.); asep044@brin.go.id (A.B.); 3Department of Pharmacy, School of Medicine and Surgery, University of Naples Federico II, Via D. Montesano 49, 80131 Napoli, Italy; g.chianese@unina.it

**Keywords:** *Andrographis paniculata*, traditional Chinese medicine, andrographolide, anticancer, antihypertensive, anti-inflammatory, antioxidant

## Abstract

*Andrographis paniculata* is widely used as a traditional medicine in Asian countries. It has been classified as a safe and non-toxic medicine by traditional Chinese medicine. The investigation of the biological activities of *A. paniculata* is still focused on the crude extract and isolation of its main active compound, andrographolide, and its derivatives. However, the use of andrographolide alone has been shown to exacerbate unwanted effects. This highlights the importance of developing a fraction of *A. paniculata* with enhanced efficacy as an herbal-based medicine. In this study, the extraction and fractionation of *A. paniculata*, followed by quantitative analysis using high-performance liquid chromatography coupled with a DAD detector, were established to quantify the andrographolide and its derivative in each fraction. Biological activities, such as antioxidant, anticancer, antihypertensive, and anti-inflammatory activities, were evaluated to study their correlations with the quantification of active substances of *A. paniculata* extract and its fractions. The 50% methanolic fraction of *A. paniculata* exhibited the best cytotoxic activities against CACO-2 cells, as well as the best anti-inflammatory and antihypertensive activities compared to other extracts. The 50% methanolic fraction also displayed the highest quantification of its main active compound, andrographolide, and its derivatives, 14-deoxy-11,12-didehydroandrographolide, neoandrographolide, and andrograpanin, among others.

## 1. Introduction

*Andrographis paniculata* (Burm.f.) Nees is a plant of the Acanthaceae family and is widely distributed in tropical areas such as Asia. *A. paniculata* is one of the most important medicinal plants in the Unani and Ayurvedic systems of medicine, which are the oldest known medicinal systems. This plant has been used in herbal medicines to treat various infectious and degenerative diseases [1].

The biological activities of *A. paniculata* are known to correlate with diterpene lactone compounds. The lactone groups are the cyclic ester of organic acid with the most stable structure; they are five-membered (gamma lactone). They are known to have various biological activities including anticancer, antifungal, and antibacterial [2,3,4,5,6], and are found in andrographolide and its derivatives, including 14-deoxy-11,12-didehydroandrographolide, neoandrographolide, and andrograpanin (Figure 1) [7]. Andrographolide has been widely studied, including, for example, for its potential medicinal value as a drug for various diseases owing to its anticancer [8], antimicrobial [9,10], antioxidant [11], anti-inflammatory [12], antidiabetic [13], and antiviral [14] properties. Meanwhile, 14-deoxy-11,12-didehydroandrographolide has shown antifungal, antiviral, and anticancer [7] activity. In addition, neoandrographolide is reported to exhibit antiviral, anti-inflammatory, and hepatoprotective activity [15,16].

*A. paniculata* has been classified by traditional Chinese medicine as a safe and non-toxic medicine [17]. To date, the investigation of the biological and pharmacological activities of *A. paniculata* is still focused on the crude extract and isolation of andrographolide and its derivatives. Several studies have examined the extraction, separation, isolation, and quantification of its bioactive compounds from *A. paniculata.* For instance, the extraction process uses various types of organic solvents and various types of extraction techniques, such as microwave and ultrasonic extraction, to obtain high yields of andrographolide and its derivatives to enhance its biological activity. Despite comparisons of the standardization for crude extracts using the major metabolites such as andrographolide, no rigorous study has aimed to develop the quantitative multi-compounds of *A. paniculata* using HPLC-DAD and the correlation with its biological activities. Multi-compound analyses of andrographolide and its derivatives from extracts and fractions of *A. paniculata* using a gradient elution system to compare the biological activity of each fraction can lead to the development of specially targeted herbal medicine. Highly sensitive, gradient compatible, and easy-to-use DAD detectors make multi-compound analysis more feasible. Therefore, in this study, we analyzed the quantification of the four main constituents of *A. paniculata*—namely, andrographolide, 14-deoxy-11,12-didehydroandrographolide, neoandrographolide, and andrograpanin—in crude methanol extract and fractions using quantitative multi-components analysis by HPLC-DAD according to their related biological activities, such as antioxidant, anticancer, antihypertensive, and anti-inflammatory activity.

## 2. Results and Discussion

### 2.1. Fractionation and Diterpenoid Quantification of A. paniculata

*A. paniculata* has often been reported to contain diterpene lactones, such as andrographolide, as its primary compound (Figure 1). The content of this plant and its derivative composition varies depending on many factors, such as the agricultural location, cultivation system, harvesting period, and post-harvesting process [18]. These variations could affect the obtained extracts’ quality and biological activities. Semi-purification using solid phase extraction (SPE) allows the grouping of compounds based on their polarity level to obtain specific extracts containing andrographolide and its derivatives. Plant compounds usually contain phenolics, diterpenoids, and flavonoids, as well as organic acids [19,20]. However, in *A. paniculata*, the active compound comes from the diterpenoid group, namely andrographolide and its derivatives. Elution carried out on SPE by various solvent compositions separates these chemical groups into specific fractions [21]. However, some non-polar compounds from the crude extract are retained in the column during the fractionation process. This can be seen after flushing the SPE cartridge using n-hexane, which leaves dark color in the column, which means the SPE fractionation process eliminates some non-polar compounds. Thus, separation using SPE becomes important for obtaining extracts rich in andrographolides and its derivatives.

The quantification method for andrographolide and its derivatives is essential to developing a standardized herbal for *A. paniculata.* In this work, we used HPLC equipped with a DAD detector to quantify the diterpene lactone content in *A. paniculata* extract and fractions. This HPLC method provides an efficient separation system and elution time to detect and quantify different components, including andrographolide and its derivatives.

Figure 2 shows the HPLC chromatogram of several diterpene lactones contained in *A. paniculata.* Under our chromatographic analysis conditions, andrographolide, 14-deoxy-11,12-didehydroandrographolide, neoandrographolide, and andrograpanin were effectively separated and observed at 8.3, 20.4, 22.3, and 37.1 min, respectively. The coefficient determination (R^2^) of the calibration curve of each compound had good fitness (0.9947–0.9972) (Table 1), indicating the reliability of the method. The peaks of compounds were also clearly separated in the chromatogram of the crude extract. As a result, the main compound in the extract of *A. paniculata* was andrographolide, followed by neoandrographolide, 14-deoxy-11,12-didehydroandrographolide, and andrograpanin.

Fractionation was carried out to determine the effectiveness of the applied method in separating the diterpenoid compounds. Elution was conducted by adjusting the polarity of methanol and water. In the first step, the separation was ineffective since the fraction still contained components similar to the crude extract, except andrograpanin (Table 1). In the second step of fractionation, F2 contained the highest content of the 4 diterpene lactones, with andrographolide, 14-deoxy-11,12-didehydroandrographolide, neoandrographolide, and andrograpanin levels of 85.65, 49.1, 110.77, and 17.39 mg g^−1^ sample, respectively. Meanwhile, the third step of fractionation (F3) only contained andrographolide at a concentration of 12.16 mg g^−1^. These results indicate that most of the andrographolide and its derivatives were concentrated in the second fractionation step. The quantity of andrographolide and its derivatives contained in each fraction might lead to the specific biological activity of each fraction [22,23]. These results indicate the use of extracts and fractions of *A. paniculata* as standardized herbal medicine ingredients with specific biological properties.

### 2.2. Total Phenolic and Flavonoid Content

A chromatographic analysis of the methanolic crude extract of *A. paniculata* indicated the presence of compounds other than andrographolide, as indicated by the observed peaks. Most notably, the plant’s extracts contained phenolic- and flavonoid-based compounds. These two compounds’ groups were determined for the crude extract and fractions of *A. paniculata* in order to obtain more comparative results (Table 2). As expected, the methanolic crude extract showed high phenolic content, with a value of 30.68 µg GAE mg^−1^ d.w. The lowest phenolic content was observed in F3, with 16.81 µg GAE mg^−1^ d.w. Herein, decreasing the polarity of the solvent during the SPE column could provide the lowest content of phenolic compounds in the less-polar fractions. On the other hand, the total flavonoid content of the extract and fractions of *A. paniculata* showed the highest value in F3, followed by ME, F2, and F1, with values of 147.77, 33.50, 19.57, and 14.68 µg QE mg^−1^ d.w., respectively. The flavonoid compound is a semi-polar compound that a semi-polar solvent will effectively extract. Hence, the low polarity of the eluent in isolating F3 seems to be more appropriate for extracting flavonoids when compared to F2 and F1, which used 50% and 10% of methanol and water, respectively. The use of semi-polar solvents such as alcohols for the extraction of *A. paniculata* generates higher levels of phenolics and flavonoids than the use of polar solvents (e.g., water) or non-polar solvents (e.g., hexane and chloroform). Alcohols, especially methanol, efficiently penetrate cells, causing osmotic changes and breaking cell walls, thus releasing all metabolites from the cells [24]. As such, the phenolics and flavonoids contained in the cells should be released. According to several reports, six main flavonoids have been successfully isolated from the ethyl acetate soluble fraction of ethanol or the methanol extract of *A. paniculata* [24,25,26].

The quantification of the diterpenoid content in the extract and fraction of *A. paniculata* using HPLC revealed that F2 contained the most diterpenoid compounds. This result supports a study conducted by Chia et al. [27]. In general, the content of diterpenoids is semi-polar compounds that are eluted with a semi-polar solvent based on the principle of like dissolves like. F1 is the most polar fraction and is dominated by phenolic group compounds. Meanwhile, F3 is more non-polar than the other fractions and, thus, is dominated by flavonoid compounds.

### 2.3. Antioxidant Activity

Antioxidant activity was evaluated using ABTS and FRAP methods. Based on various mechanisms, the antioxidant capacity can be divided into two main categories: hydrogen atom transfer (HAT) and single electron transfer (SET). FRAP involves SET reaction-based assay radicals [28], while ABTS involves a mixed-mode assay (HAT/SET) based on the elimination of stable chromophores [29]. The antioxidant capacities of the crude extract and fractions of *A. paniculata* are displayed in Table 2. The extract showed a higher antioxidant capacity (both in ABTS and FRAP) than F1, F2, and F3. This is because of the high content of phytochemicals in the crude extract, including the four diterpene lactones and other phenolic compounds of interest in the present study. F3 displayed the lowest antioxidant capacity among the fractions, resulting in its lower phytochemical content. Although this fraction showed high flavonoid content, this finding indicates that the low variety of diterpenoid lactones in F3 might also lead to its low antioxidant activity.

The high diterpenoid content in *A. paniculata* extract contributed to its increased antioxidant capacity. These results align with those presented by Akouah et al. [23], who found that the MeOH extract of *A. paniculata* contained 21.50 mg g^−1^ andrographolide and 13.30 mg g^−1^ 14-deoxy-11,12-didehydroandrographolide and produced 53.82% free radical scavenging activity. Meanwhile, they reported that the aqueous extracts of *A. paniculata* containing 0.80 mg g^−1^ andrographolide and 0.70 mg g^−1^ 14-deoxy-11,12-didehydroandrographolide produced only 28.77% free radical scavenging activity. It should be noted that the antioxidant capacity of *A. paniculata* extract was affected not only by its diterpenoid lactone, as the total phenolic content also greatly affects its antioxidant capacity. Rao and Rathod [23] explained that the t-butanol extract of *A. paniculata* with a TPC of 9.05 mg GAE g^−1^ d.w. had a higher antioxidant capacity (IC_50_ = 52.10 µg mL^−1^) than the aqueous extract (TPC = 2.04 mg GAE g^−1^; IC_50_ = 418.51 µg mL^−1^). In the antioxidant assay, the activity of a single compound was not always higher than that of the crude extract. The single compounds of andrographolide and 14-deoxy,11,12 didehydroandrographolide had lower radical scavenging activity than the methanol extract of *A. paniculata* [30]. Semi-purification (fractionation) can increase the specificity of *A. paniculata* used as standardized herbal, especially as antioxidants, because different fractions have different bioactive compounds and exhibit different biological activities.

### 2.4. Cellular Nitric Oxide (NO) Activity

Antioxidants reduce oxidative stress in the body. Some reports have shown that oxidative stress is correlated with several inflammation diseases, including strokes, heart attacks, and heart failure. Considering its antioxidant characteristics, the anti-inflammatory activity of *A. paniculata* was also examined using nitric oxide (NO) production via lipopolysaccharide (LPS)-induced in RAW 264.7 cells. When stimulated with LPS, these cells induce NO production to begin an inflammation process. Thus, samples with strong antioxidant activity commonly suppress NO production.

In this study, F2 presented the highest inhibition of NO production, with a percentage of inhibition of 98.36%. This activity is also correlated with the high antioxidant activity of F2. Considering the sufficient levels of andrographolide, 14-deoxy-11,12-didehydroandrographolide, neoandrographolide, and andrograpanin in F2, it could be speculated that these components are responsible for the potential biological anti-inflammatory activity as observed in the mechanism of inhibiting NO production. Previous studies have shown that andrographolide suppresses NO production and reduces the expression of an inducible isoform of the NO synthase enzyme, which is responsible for NO production [24,31]. Andrographolide derivatives, 14-deoxy-11,12-didehydroandrographolide, neoandrographolide, and andrograpanin are also reported to inhibit NO production in LPS-induced murine macrophages in a concentration-dependent manner.

Meanwhile, 14-deoxy-11,12-didehydroandrographolide exhibited NO inhibition with an IC_50_ value of 94.12 ± 4.79 µM [32]. Neoandrographolide inhibited NO production with an IC_50_ value of more than 100 µM [33,34]. Similar to neoandrograpaholide, andrograpanin inhibited NO production in a concentration-dependent manner, but it was found to have an IC_50_ value of more than 100 µM [35]. Therefore, the presence of andrographolide and its derivatives in *A. paniculata* samples significantly affects NO suppression in LPS-induced murine macrophages [22]. It should be noted that the activity of other fractions and extracts of *A. paniculata* that inhibit NO production was still higher than that of the positive control of dexamethasone, which only inhibited 84.82% of NO production (Table 3).

### 2.5. Cytotoxic Activity

Additional insights were gained by investigating the cytotoxic activity of *A. paniculata* extract and fractions in CACO-2 cell lines using MTT assay. The assay is based on the reduction of MTT reagent by mitochondrial dehydrogenase from the introduction of active metabolic cells into the formazan (blue form). Figure 3 illustrates CACO-2 cell line viability before and after treatment with each *A. paniculata* extract and fraction, as observed under 40 times magnification using an inverted microscope. Figure 4 summarizes the cytotoxic activity of each sample on CACO-2 cell lines. In general, the IC_50_ values of the samples were less than 100 µg mL^−1^, except for F3, which had values of up to 145.10 µg mL^−1^. The highest CACO-2 cell growth inhibition activity was exhibited by F2, with an IC_50_ value of 32.46 µg mL^−1^, followed by ME (63.89 µg mL^−1^) and F1 (61.04 µg mL^−1^). These findings are consistent with the high anti-inflammatory activity test results of F2, which has the highest levels of andrographolide, 14-deoxy-11,12-didehydroandrographolide, neoandrographolide, and andrograpanin. Moreover, the low activity of F3 is in line with the small amount of andrographolide and the absence of 14-deoxy-11,12-didehydroandrographolide, neoandrographolide, and andrograpanin.

### 2.6. Angiotensin-Converting Enzyme (ACE) Inhibition Activity

Angiotensin-converting enzymes (ACEs) are membrane-bound proteins that play an essential role in regulating blood pressure and the electrolyte balance system [36]. The inhibition of ACEs can regulate blood pressure protection in arteries. Moreover, they are associated with inflammation. Based on the results of an ACE inhibition test, F1 and F2 showed the highest inhibition (up to 94%), followed by ME (81%) and F3 (21%) (Figure 5). The inhibitory activity of the ACE was influenced by the content of metabolites in each extract and fraction, as fractions with a high content of secondary metabolites showed high activity. An exception was that ME had a higher secondary metabolite content than F1 but exhibited less inhibitory activity. This may be due to the phenolic and flavonoid content of ME, which may be antagonistic in inhibiting ACE activity.

The high content of andrographolide, 14-deoxy-11,12-didehydroandrographolide, neoandrographolide, and andrograpanin of F2 corresponds to this fraction’s high level of activity as an anticancer agent with an IC_50_ value of 32.46 µg mL^−1^ for cytotoxicity to CACO-2; the suppression of 98.36% NO production in RAW 264.7; and the inhibition of 94% ACE activity (which is the highest among all fractions). However, this activity is different from antioxidant activity. The ABTS and FRAP assay results show that ME has more potential as an antioxidant agent than the three fractions owing to its high phenolic content.

The correlation between each extract’s diterpenoid, phenolic, and flavonoid content and its fractions on its biological activity can be considered when manufacturing standardized herbal medicines. In addition, insights into the correlation can help increase the efficacy and specificity of developed herbal medicines. Efficacy and specificity can be increased by further separating the extracts used as ingredients in herbal medicines. For example, after the fractionation process on ME, F2 was obtained, which exhibits more anticancer, antihypertensive, and anti-inflammatory activity than ME, F1, and F3. However, the antioxidant activity of F1 was almost the same as that of ME and higher than that of F2 and F3. After separation, specific fractions were identified as antioxidant, anticancer, anti-inflammatory, and antihypertensive agents.

## 3. Materials and Methods

### 3.1. Materials

Reagents and standards were obtained from Sigma-Aldrich (Sigma-Aldrich, St. Louis, MO, USA), including andrographolide, 14-deoxy-11,12-didehydroandrographolide, neoandrographolide, andrographanin, Folin-Ciocalteu reagent, ABTS (2,2′-Azino-bis-3ethylbenzothiazoline-6-sulfonic acid), TPTZ (2,4,6-tripyridyl-s-triazine), K_2_S_2_O_8_, FeCl_3_, CH_3_COONa, Trolox (6-hydroxy-2,5,7,8-tetramethylchroman-2-carboxylic acid), MTT (3-(4 5-dimethyltiazole-2-yl)-2,5-diphenyltetrazolium bromide), Dulbecco’s Modified Eagle Medium (DMEM), fetal bovine serum (FBS), L-glutamine, penicillin, streptomycin, methanol (MeOH), ethanol, dichloromethane, ethyl acetate, and n-hexane. All chemical reagents were of pro-analytical and chromatographical grade. The human colorectal adenocarcinoma (CACO-2 ECACC 86010202) cell line and murine macrophage cell line (RAW 264.7 cells ECACC 91062702) were used for anticancer and anti-inflammation assays, respectively. These were obtained from the European Collection of Authenticated Cell Cultures (ECACC). *A. paniculata* was obtained from the Center for Research and Development hof Medicinal Plants and Traditional Medicine (B2P2TOOT), National Institute of Health Research and Development, Indonesian Ministry of Health, Tawangmangu, Central Java, Indonesia.

### 3.2. Methods

#### 3.2.1. Sample Preparation and Extraction Method

*A. paniculata* was harvested after a cultivation period of eight months and dried in the oven. The aerial parts, including leaves and roots, were ground and powdered using a pulverizer mill. Afterward, 200 g of powder was extracted with 850 mL MeOH at room temperature for 24 h. The mixture was then filtrated, and the spent biomass was re-extracted with 850 mL MeOH for 24 h. The extraction process was conducted three times. The filtrates were collected and concentrated using a vacuum evaporator (Buchi Rotavapor R-300 Series, Swiss) and lyophilized (Buchi Lyovapor L-200, Swiss) to obtain the extract (19.25 g). Then, about 1 g of the dried extract was fractionated using SPE with ODS C-18 Bond Elut (6 mL of cartridge volume and 500 mg of sorbent) cartridge. A gradient system was applied by decreasing the polarity of the eluents from H_2_O/MeOH (9:1), which was marked as F1; H_2_O/MeOH (1:1), which was marked as F2; and MeOH 100%, which was marked as F3. Then, the cartridge was flushed with n-hexane for reuse up to two times. A total of 4 g of dried sample was fractionated, and the yields of each fraction obtained (F1, F2, and F3) were 83.2%, 5.1%, and 8.2%, respectively.

#### 3.2.2. HPLC Analysis and Quantification of Andrographolide and Its Derivatives

The analysis and quantification of andrographolide and its derivatives were performed using the method described by Dalawai et al. [37], with slight modification. An HPLC system (UFLC Shimadzu, Kyoto, Japan) equipped with a photodiode array detector and analytical column Thermo Hypersil ODS C18 (250 × 4.6 mm, 5 µm) was used to quantify andrographolide and its derivatives. Water and methanol were used as a mobile phase for the chromatographic conditions of MeOH/H_2_O (1:1) for 25 min and MeOH/H_2_O (3:2) for 15 min at a flow rate of 1 mL/min^−1^. Before the analysis, the liquid sample of the extract or fraction was filtered using a 0.45-µm syringe filter. About 20 µL of the sample was injected, and the chromatogram was recorded at 210 nm. The quantification of andrographolide and its derivatives was carried out by constructing a combined calibration curve of the 4 standard compounds with a linearity range of 5 to 500 ppm. The area of crude extract and fractions was interpreted into a calibration curve.

#### 3.2.3. Total Phenolic Content (TPC) by Folin-Ciocalteu Reagent Assay

The Folin-Ciocalteu method determined the TPC value following Sekhon-Loodu and Rupasinghe [38]. First, 100 µL of Folin-Ciocalteu reagent (0.2 N) was added to each well of a 96-well plate and mixed with 20 µL of the sample solution (1 mg mL^−1^). After 5 min, 80 µL of a Na_2_CO_3_ solution (7.5%) was added to each well and incubated for 2 h at room temperature in dark conditions. The absorbance was then measured at 760 nm using the microplate reader (Tecan Infinite 200 Pro, USA). A similar procedure was applied to determine gallic acid as the standard solution with concentrations of 6.25, 12.50, 25.00, 50.00, and 100.00 µg ml^−1^. The TPC value was expressed as the weight of gallic acid equivalent (GAE) per g of dry extract (µg GAE g^−1^ dry weight).

#### 3.2.4. Total Flavonoid Content (TFC) by Aluminum Chloride Colorimetric Assay

The TFC analysis was performed using the assay protocol described by Chatatikun et al. [39]. First, 10 µL of an AlCl_3_ solution (10%) and 50 µL of each sample were added to a 96-well plate. Then, 10 µL of a CH_3_COONa solution (1 M) and 150 µL of ethanol were added to the mixtures. Afterward, the mixed solution was incubated and shaken for 40 min at room temperature in dark conditions. Then, the absorbance was measured at 415 nm using a microplate reader (Tecan Infinite 200 Pro). The total flavonoid content was presented as mg quercetin equivalent (QE) per g dry extract (µg QE g^−1^ dry weight). Various concentrations of the quercetin solution (i.e., 0.5, 1, 2.5, 5, 10, and 25 µg mL^−1^) were used as a standard.

#### 3.2.5. 2,2′-Azino-bis-3ethylbenzothiazoline-6-sulfonic Acid (ABTS) Assay

The ABTS scavenging activity was examined using the method described by Xiang et al. [40], with some modifications. The ABTS solution was prepared by mixing 5 mL of ABTS (7.4 mM) with 88 µL of K_2_S_2_O_8_ (2.6 mM). The stock solution was incubated for 16 h at room temperature in dark conditions to produce ABTS+. The ABTS+ solution was diluted with water to obtain an initial absorbance of around 0.7 at 734 nm. Subsequently, 10-µL samples were added to a 96-well plate and mixed with 190 µL of the ABTS stock solution. Then, the mixed solution was incubated in the dark at room temperature for six minutes. The absorbance was measured at 734 nm using a microplate reader (Tecan Infinite 200 Pro). The scavenging effect was expressed as mM Trolox equivalent per g dry weight of the extract. The standard curve of the Trolox solution was prepared from various concentrations ranging from 100 to 2000 μmmol L^−1^.

#### 3.2.6. Ferric Reducing-Antioxidant Power (FRAP) Assay

FRAP assays were carried out according to the protocol described by Li et al. [41], with slight modifications. A FRAP solution was prepared by mixing a sodium acetate solution (300 mM, pH 3.6) and a TPTZ solution (10 mM) in a solution of 40 mM HCl and FeCl_3_ (20 mM) at a 10:1:1 ratio (*v*/*v*). A fresh 150 µL FRAP solution was mixed with a 50-µL sample or standard in a 96-well plate and incubated at 37 °C for four minutes. The absorbance was measured at 593 nm using a microplate reader (Tecan Infinite 200 Pro). The FRAP results were converted into mM of Trolox (6-hydroxy-2,5,7,8-tetramethylchroman-2-carboxylic acid) equivalents per mg (mM TE mg^−1^ d.w.) of samples using the standard curve, plotted at different concentrations of the standard ranging from 0.06 to 1.00 mM.

#### 3.2.7. Angiotensin-Converting Enzyme (ACE) Inhibition Assay

The ACE inhibitory activity was carried out based on the method presented by Hettihewa et al. [42], with slight modifications. The reaction mixtures contained 30 µL of ACE 2.5 mU, 9 µL of a sodium borate buffer (pH 8.3), and 150 µL of hippuryl-histidyl-leucine 0.78 mM (HHL), which were added into a serial concentration of samples (21 µL) in a microtube. The mixtures were incubated via shaker incubation at 37 °C for 1 h. Afterward, 150 µL of NaOH 0.35 M was added to stop the enzyme activity. The formation of hippuric acid from the enzymatic hydrolysis of HHL by the enzyme was measured by a microplate reader at 450 nm using a microplate reader (Tecan Infinite 200 Pro). The following equation was used to calculate the ACE inhibition:(1)$$\mathrm{ACE}\;\mathrm{Inhibition}\;\left(\%\right)=\frac{\left(B-A\right)}{\left(B-C\right)}\times 100,$$
where B is the absorbance of positive control; A is the absorbance of the solution containing ACE, HHL, and the inhibitor component; and C is the absorbance of the negative control (ACE and HHL without the inhibitor) [43].

#### 3.2.8. Cell Culture

The murine macrophage cell line (RAW 264.7 cells ECACC 91062702) and human colorectal adenocarcinoma (CACO-2 ECACC 86010202) cell line were cultured with DMEM containing 10% FBS, 2 mmol L^−1^ L-glutamine, 100 U mL^−1^ penicillin, and 10 μg mL^−1^ streptomycin. These cell lines were maintained at 37 °C in a humidified atmosphere of 5% CO_2_ (CO_2_ incubator, Thermo Scientific, Waltham, MA, USA).

#### 3.2.9. Cellular NO Production and Quantification

RAW 264.7 cells (5 × 104 well^−1^) were cultured in a 96-well plate and pretreated with extracts and fractions of *A. paniculata* for 1 h before being stimulated with LPS for 20 h. Cellular NO production was determined by assaying the NO^2−^ concentration in the culture supernatant. In this process, 100 μL of culture supernatant was mixed with an equal volume of Griess reagent (1% sulfanilamide and 0.1% N-[naphthyl]ethylene diamine dihydrochloride in 2.5% H_3_PO_4_). The mixture was incubated at room temperature for 10 min. Afterward, the absorbance was measured at 540 nm using a microplate reader (Cytation 5, BioTek, Winooski, VT, USA). Nitrite concentration was calculated from a NaNO_2_ standard curve.

#### 3.2.10. Cytotoxic Screening

Cytotoxic screening was performed using the MTT assay method. CACO-2 cells were plated in a 5 × 104 cells well^−1^, followed by incubation for 24 h at 37 °C. After this incubation period, cells were treated with methanolic extract, F1, F2, and F3 of *A. paniculata* and dimethyl sulfoxide (as a positive control) and incubated for 48 h. After that, the medium was replaced, and MTT reagents (0.5 mg mL^−1^) containing the new medium were added to the 96-well plate. Incubation was continued for 4 h at 37 °C in a CO_2_ incubator (Thermo Fischer). Absorbance was measured using a microplate reader (Cytation 5, BioTek) at 560 nm. Log concentration vs. normalized absorbance was plotted to obtain the IC_50_ value of each extract.

#### 3.2.11. Statistical Analysis

All data were collected in triplicate and expressed as mean ± SD. The differences between various data were analyzed with a single factor ANOVA with a 95% confidence level. Microsoft Excel 365 with additional data analysis plugins was used to run the single-factor ANOVA.

## 4. Conclusions

Different extracts and fractions of *A. paniculata* exhibited different biological activities depending on the concentration of andrographolide and its derivatives. The high content of andrographolide, 14-deoxy-11,12-didehydroandrographolide, neoandrographolide, and andrograpanin in F2 resulted in the highest levels of biological activity, such as cytotoxicity, against CACO-2 cells (IC_50_ = 32.46 g mL^−1^), inhibiting NO production in RAW 264.7 cells (inhibition = 98.36%), and inhibiting ACE activity (inhibition = 94.19%). On the other hand, the antioxidant capacity was strongly influenced by not only the content of andrographolide and its derivatives but also the phenolic content. Therefore, the highest level of antioxidant activity was exhibited by ME with Trolox equivalent values of 0.35 and 0.47 mmol TE g^−1^ dry weight.

The relationship between the content of diterpenoids, phenolics, and flavonoids in the extract and the fractions of *A. paniculata*, owing to their biological activity, can be used as a reference in the development of standardized herbal medicines.

## Figures and Tables

**Figure 1 plants-12-01220-f001:**
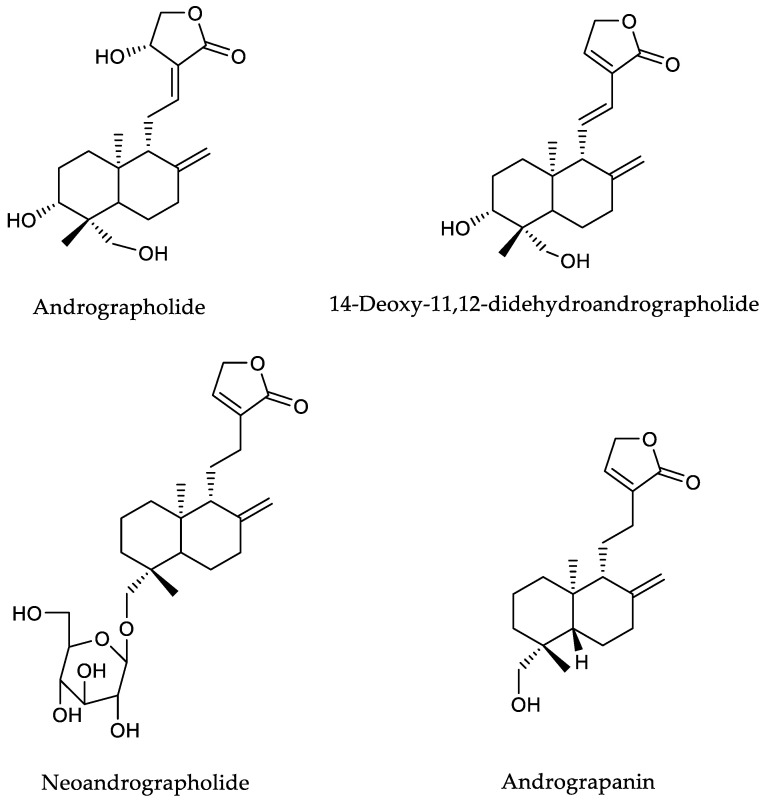
Chemical structures of four diterpene lactones from *A. paniculata*.

**Figure 2 plants-12-01220-f002:**
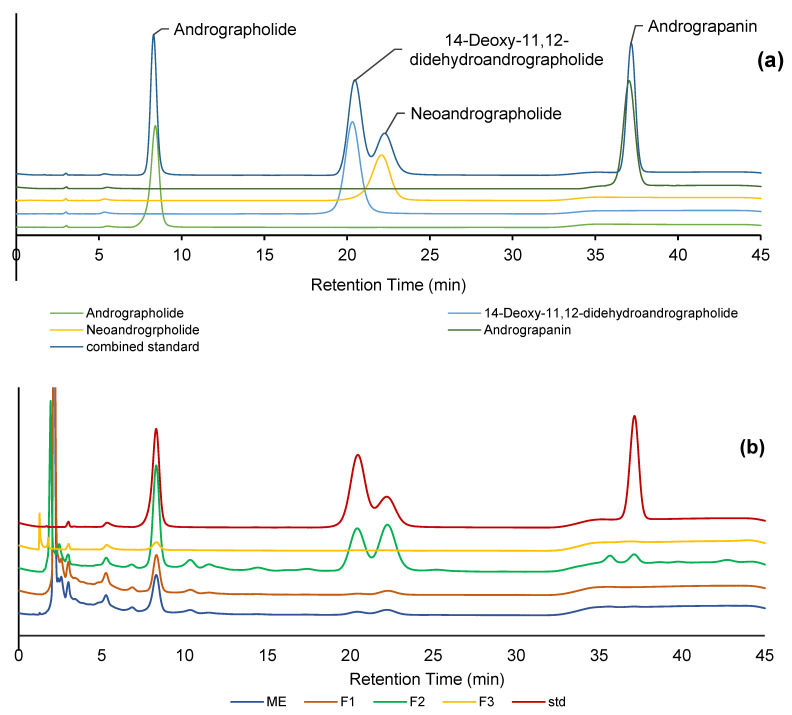
(**a**) An HPLC chromatogram of the standard compounds of andrographolide and its derivatives and (**b**) an HPLC chromatogram of extract and fractions compared with the standard compounds. ME, F1, F2, F3, and std are methanolic crude extract, fraction 1, fraction 2, fraction 3, and the standard solution. The details of the chromatographic analysis can be found in the experimental method.

**Figure 3 plants-12-01220-f003:**
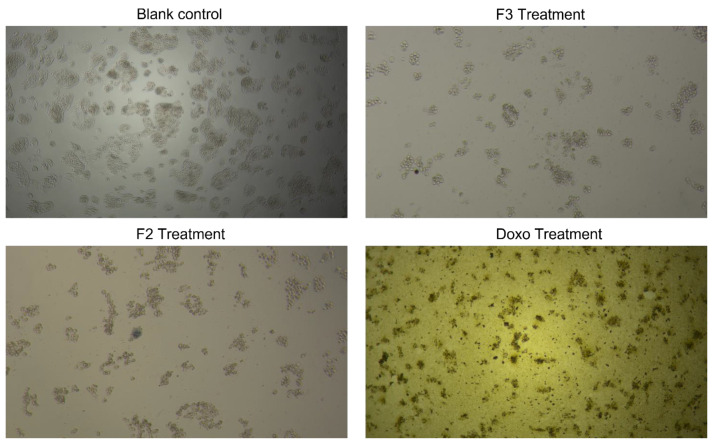
Morphological changes in CACO-2 cell lines when treated with F2 and F3 and doxorubicin when compared to a blank control (untreated cell).

**Figure 4 plants-12-01220-f004:**
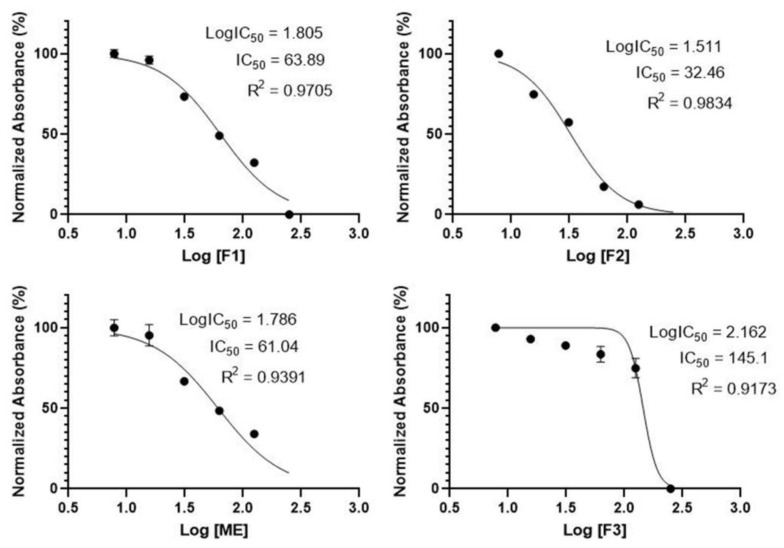
Cytotoxic activity of *Andrographis paniculata* extract and its fractions expressed as 50% inhibition concentration (IC_50_) values. F2, which contains the highest levels of the four investigated diterpene lactone compounds, shows the lowest IC_50_ value, followed by ME and F1. F3, which contains only a small amount of andrographolide, has the highest IC_50_ value.

**Figure 5 plants-12-01220-f005:**
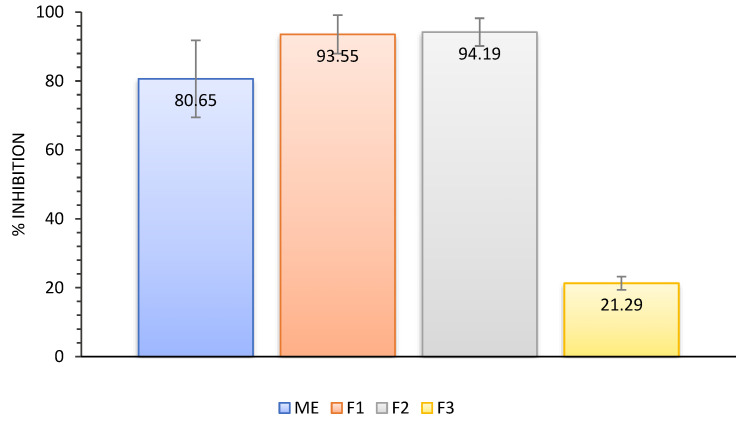
Inhibition of angiotensin-converting enzyme (ACE) activity by *A. paniculata* extract and fractions that showed antihypertensive activity (expressed as percentages) (*p* < 0.05).

**Table 1 plants-12-01220-t001:** Quantification of four diterpene lactones from crude methanol extract and the SPE fraction of *A. paniculata*.

Compound	Samples (mg g^−1^)			
	ME	F1	F2	F3
Andrographolide	38.15	31.45	85.65	12.16
14-deoxy-11,12-didehydroandrographolide	8.60	6.43	49.19	-
Neoandrographolide	19.87	16.86	110.77	-
Andrograpanin	4.98	-	17.39	-

ME = methanolic crude extract; F1 = fraction from the 10% methanol step; F2 = fraction from the 50% methanol step; F3 = fraction from the 100% methanol step.

**Table 2 plants-12-01220-t002:** Antioxidant capacity and total phenolic and flavonoid content of the extract and fractions obtained from *Andrographis paniculata* (*p* < 0.05).

Extract	Antioxidant Capacity (mmol TE g^−1^ d.w.)		TPC (µg GAE g^−1^ d.w.)	TFC (µg QE g^−1^ d.w.)
	ABTS	FRAP		
ME	0.35 ± 0.01	0.47 ± 0.01	30.68 ± 2.34	33.50 ± 4.57
F1	0.23 ± 0.01	0.25 ± 0.02	30.21 ± 1.13	14.68 ± 5.11
F2	0.22 ± 0.00	0.08 ± 0.00	20.02 ± 2.30	19.57 ± 4.17
F3	0.12 ± 0.00	0.06 ± 0.01	16.81 ± 0.21	147.77 ± 5.30

**Table 3 plants-12-01220-t003:** Methanol extract and fractions 1, 2, and 3 of *Andrographis paniculata* suppressed the production of nitric oxide (NO) in RAW 264.7 cells showing anti-inflammatory activity (*p <* 0.05).

Sample	NO Production (µM)	% Inhibition
Dexamethasone 10 ppm	3.80 ± 0.21	84.82
ME	0.81 ± 0.18	96.78
F1	1.33 ± 0.46	94.70
F2	0.41 ± 0.11	98.36
F3	2.18 ± 0.24	91.29

## Data Availability

Not applicable.

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
