# Peer review of "Evaluations of Andrographolide-Rich Fractions of Andrographis paniculata with Enhanced Potential Antioxidant, Anticancer, Antihypertensive, and Anti-Inflammatory Activities"

_plants, 2023, doi:10.3390/plants12061220_

Round 1

Reviewer 1 Report

The manuscript deals with the evaluation of some biological properties of the crude methanolic extract of A. paniculata and its fractions.
In my opinion, the paper needs to be revised by a native English speaking expert before publication.
In addition, some more information is needed in the Mat & methods section.
Which are the dimensions/volumes of the SPE cartridges used for fractionation? Was a single SPE cartridge or several used? How many replicates of the fractionation process were performed?
The description of the quantitative method used for andrographolide and its derivatives with the corresponding performance specification is missing.
In my opinion, the fractionation process simply eliminates some compounds that are not detectable with HPLC-DAD instrument and enriches them in fraction F2: this should be discussed
The sentence in lines 181-182 is not clear: why can the semi-purification step increase the specificity of the A. paniculata extract as antioxidant? The crude methanolic extract is more active than the fractions.
There are also some technical problems in citing the references in the text. The message: "Error! Reference source not found", is present for some of them, as in lines 42, 215, 225, 249

Reviewer 2 Report

The authors should revise the structures of some compounds prior preparation and add a couple of reference

Round 2

Reviewer 1 Report

The manuscript has been revised taking into account the reviewer's suggestion,  therefore it can now be  accepted for publication in Plants

Reviewer 2 Report

After seeing the changes made by the authors I consider that should be published in the present form